

# Bidirectional k-nearest neighbor spatial crowdsourcing allocation protocol based on edge computing

Jing Zhang[1], Qian Ding[1], Biao Li[1] and Xiucai Ye[2]

[1] School of Computer Science and Mathematics, Fujian Provincial Key Laboratory of Big Data Mining and Applications, Fujian University of Technology, Fuzhou, Fujian, China

[2] Department of Computer Science, University of Tsukuba, Tsukuba, Japan

## ABSTRACT

Spatial crowdsourcing refers to the allocation of crowdsourcing workers to each task based on location information. K-nearest neighbor technology has been widely applied in crowdsourcing applications for crowdsourcing allocation. However, there are still several issues need to be stressed. Most of the existing spatial crowdsourcing allocation schemes operate on a centralized framework, resulting in low efficiency of crowdsourcing allocation. In addition, these spatial crowdsourcing allocation schemes are one-way allocation, that is, the suitable matching objects for each task can be queried from the set of crowdsourcing workers, but cannot query in reverse. In this article, a bidirectional k-nearest neighbor spatial crowdsourcing allocation protocol based on edge computing (BKNN-CAP) is proposed. Firstly, a spatial crowdsourcing task allocation framework based on edge computing (SCTAFEC) is established, which can offload all tasks to edge nodes in edge computing layer to realize parallel processing of spatio-temporal queries. Secondly, the positive k-nearest neighbor spatio-temporal query algorithm (PKNN) and reverse k-nearest neighbor spatio-temporal query algorithm (RKNN) are proposed to make the task publishers and crowdsourcing workers conduct two-way query. In addition, a road network distance calculation method is proposed to improve the accuracy of Euclidean distance in spatial query scenarios. Experimental results show that the proposed protocol has less time cost and higher matching success rate compared with other ones.

**Submitted** 11 November 2022
**Accepted** 17 January 2023
**Published** 20 February 2023

Corresponding author
Jing Zhang, jing165455@126.com

## INTRODUCTION

The development of 5G technology promotes the wide application of intelligent services, especially location-based crowdsourcing service platforms. Crowdsourcing is defined as a business model in which tasks traditionally performed by company employees are outsourced to a large group of voluntary non-specific masses in an open form (*Howe, 2006*). Nowadays, crowdsourcing has been widely applied in various fields, including environmental monitoring, medical care, Meituan, Didi Taxi, and Amazon's Turk robot (*Gummidi, Xie & Pedersen, 2019*). At the same time, everyone with smart devices on

mobile terminals can register as a worker on the crowdsourcing platform on the Internet. In this context, an emerging crowdsourcing mode of outsourcing tasks with spatial characteristics to workers on the Internet emerges, which is called spatial crowdsourcing (*Peng et al., 2022*; *Liu et al., 2022*; *Tong et al., 2020*). In the spatial crowdsourcing platform, the location-related task information is published by the task publisher to the platform server, and then the specific task assignment algorithm is used by the platform server to assign the corresponding crowdsourcing worker for the task publisher to provide service, thus forming a benign business cycle model (*Frigerio, Schenato & Bossi, 2016*; *Wang et al., 2020*; *Qawqzeh et al., 2021*; *Wu et al., 2022*).

Ride-hailing platform is a typical spatial crowdsourcing application. Due to its large user group, its service quality has attracted more and more attention (*Zhao et al., 2019*; *Chen et al., 2019*; *Bhatti, Gao & Chen, 2020*). At present, drivers are slow to receive orders and take a long time to arrive, leading to emotional anxiety and even anger among users, which is one of the biggest pain points for users of ride-hailing platforms (*Javid, Abdullah & Ali, 2022*). Therefore, in the spatial crowdsourcing platform, how to provide intelligent services while ensuring high service quality is an important social problem to be solved urgently.

The service quality of spatial crowdsourcing platform is mainly determined by the matching time efficiency, matching success rate and other indicators of crowdsourcing tasks, which are closely related to the platform's system framework and task matching algorithm. Firstly, the traditional spatial crowdsourcing platform system framework is mostly based on the central server, and each task publisher and crowdsourcing worker need to upload their location information to the central server for calculation, which greatly increases the computing overhead of the server and reduces the computational efficiency of the algorithm (*Zhang et al., 2020*). Secondly, the existing crowdsourcing task allocation algorithms are all one-way allocation, that is, they can only find the optimal crowdsourcing worker for the task publisher among multiple crowdsourcing workers, and cannot conduct two-way parallel spatio-temporal query between the task publisher and the crowdsourcing workers (*Frigerio, Schenato & Bossi, 2016*; *Yang et al., 2019*). In addition, in most of the existing crowdsourcing task allocation algorithms, Euclidean distance is used as the basis for task allocation, and the underlying road network information is ignored, resulting in low accuracy of crowdsourcing matching and seriously affecting the satisfaction of users' service experience (*Elmongui, Mokbel & Aref, 2013*; *Li et al., 2014*).

To overcome the shortcomings of traditional approaches, a bidirectional k-nearest neighbor spatial crowdsourcing allocation protocol based on edge computing (BKNN-CAP) is proposed in this article. The specific contributions of this work are as follows:

(1) A spatial crowdsourcing task allocation system framework based on edge computing (SCTAFEC) is proposed, which includes three parts: the center control layer, the edge computing layer and the crowdsourcing worker layer. With this framework, all tasks can be unloaded to the edge nodes of the edge computing layer for processing, which realizes parallel processing of spatiotemporal queries and greatly reduces the time cost of the algorithm.

(2) The positive k-nearest neighbor spatio-temporal query algorithm (PKNN) and reverse k-nearest neighbor spatio-temporal query algorithm (RKNN) are proposed. The

two algorithms can realize bidirectional parallel spatio-temporal query between task publishers and crowdsourcing workers, thus improving the efficiency of crowdsourcing task allocation.

(3) The real road network is modeled and a road network distance calculation method is proposed to improve the matching accuracy of crowdsourcing tasks.

The remainder of this article is organized as follows. The first section reviews the related work. The second section describes the problem. The third section proposes a protocol to solve the problem. The fourth section analyzes the experiments and results. At last, the current work is summarized and and the future work is prospected.

## RELATED WORK

Concerning work related to our proposal, the research on spatial crowdsourcing task allocation, edge computing and k-nearest neighbor (KNN) technology is briefly reviewed.

### Spatial crowdsourcing task allocation

Task allocation is a core issue in the field of spatial crowdsourcing, that is, how to arrange appropriate crowdsourcing workers for crowdsourcing tasks. According to the classification of problems, task allocation problems can be divided into a task matching problem and task planning problem. Task matching problems usually match crowdsourcing workers and crowdsourcing tasks one-to-one or one-to-many. In *Xing et al. (2019)*, the matching problem is combined with game theory to improve the matching satisfaction of users. *Zhang et al. (2019)* propose a reliable task allocation problem according to the reliability of workers and the employment cost. The task planning problem is to plan paths for workers to accomplish multiple tasks. In *Deng, Shahabi & Demiryurek (2013)*, the problem of planning paths for workers in offline scenarios is studied in order to accomplish as many tasks as possible. *Tao et al. (2018)* study how to maximize the total utility value (revenue) of completing tasks when workers and tasks arrive online for online scenarios.

Although these works have studied the task allocation problem in the spatial crowdsourcing platform, the large task allocation problem is studied in the centralized system framework, which is very different from the parallel processing problem model focusing on spatio-temporal queries in this article.

### Edge computing

Edge computing refers to a technology that integrates intelligent services such as network, computing and storage near the edge of the network near the object or data source (*Zhu & Xiao, 2022*). Since the service platform is at the edge of the network, compared with the traditional central cloud network, the distance between users or devices and the service platform is closer both at the physical level and the network topology level, so it can more conveniently meet the service requirements of intelligent applications and real-time services.

In recent years, edge computing technology is used to improve the efficiency of spatial crowdsourcing platform (*Xu et al., 2021*; *Xiao et al., 2020*; *Liao & Wu, 2020*). In *Xu et al. (2022)*, the edge computing is used to study a distributed spatial crowdsourcing mechanism,

which deployed some location services on edge clouds and ensured a low time delay. Based on the idea of edge computing, *Wang et al. (2021)* propose a recruitment framework based on crowdsourcing, which utilizes the power of crowdsourcing workers to provide additional communication resources and enhance communication capabilities. In this framework, intelligent network box nodes are creatively used as edge layer devices to improve the efficiency of industrial Internet based on edge computing. In *Li et al. (2019)*, a hybrid computing framework is proposed, and an intelligent resource scheduling strategy is designed to meet the real-time requirements of intelligent manufacturing supported by edge computing.

Despite the above crowdsourcing scheduling strategy based on edge computing can improve the scheduling efficiency to a certain extent. In practice, due to the limited computing power of the edge server and network bandwidth, when users in the same area send a large number of computing offload requests at the same time, resource competition will occur, which will affect the network performance of the edge computing layer. Therefore, it is a big challenge to design the corresponding task offload algorithm and give the offload strategy for the user equipment in the application of edge computing technology.

## K-nearest neighbor technology

K-nearest neighbor technology (KNN) is one of the most important techniques for spatial query. It refers to finding $k$ data objects closest to the query point $Q_i$ from a data object set $O$ (*Guan et al., 2021*). However, most of the current KNN query algorithms find $k$ nearest neighbors by Euclidean distance (*Zhu et al., 2016*), and the nearest neighbor object calculated is not the nearest to the road network, resulting in low accuracy of calculation. To solve this problem, the KNN algorithms based on road network are proposed to ensure the accuracy of spatial query results (*Abeywickrama, Cheema & Storandt, 2020*; *Chung, Hyun & Loh, 2022*; *Yang, Tang & Zhang, 2019*). However, the above algorithms are only applicable to the static scenario, that is, the query point and the data object are stationary. Based on this, the continuous k-nearest neighbor (CKNN) algorithms are proposed, which can query the k-nearest object of the corresponding query point when both the data object and the query point are moved (*Miao et al., 2020*; *Huang, Chen & Lee, 2009*). In order to further improve the efficiency of KNN query, *Bok, Park & Yoo (2019)* propose a method to effectively process CKNN query using distance relation pattern (DRP). The DRP is a list of relative coordinates sorted in ascending order by the distance between the points in a cell and other cells so that cells can be accessed sequentially. However, this method is to conduct CKNN simulation query on grid cells, which cannot be applied to the actual road network scenario.

Although KNN technology in road network has been widely used in spatial query, the existing KNN algorithms are all one-way query, that is, they can only find the KNN result of a query point $Q_i$ in the data object set $O$, but cannot find the KNN result of a data object $O_j$ from the query point set $Q$. Therefore, how to utilize it for bidirectional spatial query and combine it with task allocation problem in crowdsourcing platform is a big challenge at present.

# PROBLEM DESCRIPTION

The subsection "Motivation scenario" shows a motivating example to inspire our research. The problem definition includes the representation of basic definition and problem formalization are described in subsection "Problem definition".

## Motivation scenario

In this section, an example of the operational framework of the ride-hailing platform is introduced to illustrate the problems existing in the current spatial crowdsourcing platform. As shown in Fig. 1, the current ride-hailing operation framework is mostly centralized, which includes three parts: passengers, drivers and central server. In this framework, passengers can be seen as task publishers and drivers as crowdsourcing workers. Task requests containing their location information are posted by passengers to a central server, which assigns each passenger a corresponding driver based on a specific matching algorithm. However, in the centralized framework, all tasks need to be centrally processed by the central server, which leads to low computing resource utilization and scheduling efficiency.

In addition, the task allocation algorithm in online ride-hailing platform can be regarded as a spatial query processing process. However, the existing spatial query process mostly queries the matching results for passengers according to the Euclidean distance, and rarely considers the real urban road network structure. As a result, the spatial query results are not the drivers closest to the actual road network, which makes the service quality of the platform poor. Therefore, it is necessary to design a new platform framework and task allocation algorithm to improve the scheduling efficiency and service quality of the platform.

## Problem definition

**Definition 1 (road network, $N$).** Figure 2 shows a road network. The road network is a undirected weighted graph $N = (V, E, W)$. Where $V$ is the set of crossroads, also known as the road network node set. $E$ represents the set of sections of the road, also known as the set of edges in the network. $W$ is the set of weights of sections, each element in $W$ refers to the length of the corresponding section of road. If the edges and weights between nodes are known, it is easy to calculate the shortest path length between nodes. The road network node $v_i \in V$, $v_i = (L_{v_i}, DQ_{v_i}, DO_{v_i})$. Where $L_{v_i}$ is the position of $v_i$, $DQ_{v_i}, DO_{v_i}$ are two sets, which contain road network node $v_i$ sorting all query points and data objects in ascending order of road network distance within a certain range. In Fig. 2, the black solid dot such as $v_6$ is a road network node, $DQ_{v_6} = (Q_3, Q_5, Q_1)$. That's to say, over $K$ miles, $Q_3, Q_5$ and $Q_1$ are closest to $v_6$. Similarly, $DO_{v_6} = (O_4, O_5, O_2)$. That is, the closest data objects to $v_6$ are $O_4, O_5$ and $O_2$.

**Definition 2 (query point, $Q$).** Query point $Q_i$ refers to the user who publishes task information in the spatial crowdsourcing platform, also known as task publisher. Such as the passenger who initiates a taxi request in the taxi platform. As shown in Fig. 2, cross stars are query points in the road network. The query points set $Q = \{Q_1, Q_2, \ldots, Q_i\}$, $Q_i = (LQ_i, DV_{Q_i}, DO_{Q_i})$. Where $LQ_i$ is the current position of $Q_i$, $DV_{Q_i}, DO_{Q_i}$ are two sets,

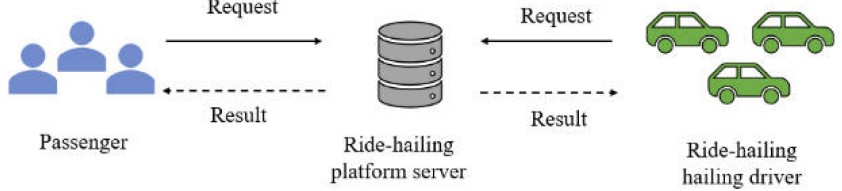

**Figure 1** A scenario showing the inefficient operation of ride-hailing platform.

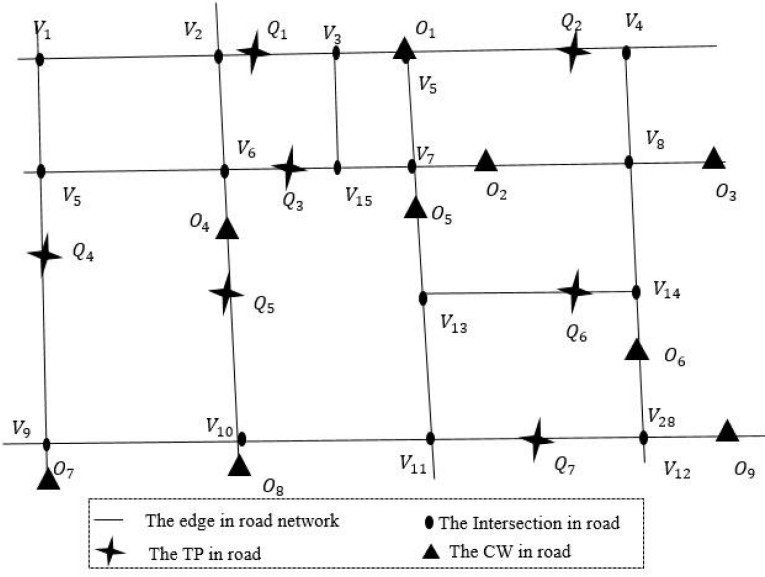

**Figure 2** An example of real road network.

which contain query point $Q_i$ sorting all network nodes and data objects in ascending order of road network distance within a certain range.

**Definition 3 (data object, $O$).** The data object $O_j$ refers to the worker who sends and processes the task request in the spatial crowdsourcing platform, also known as crowdsourcing worker. Such as the driver who completes the passenger carrying task in the taxi platform. The triangular points in Fig. 2 are the data objects in the road network, denoted by the set $O$, $O = \{O_1, O_2, \ldots, O_j\}$, $O_j = (LO_j, DV_{O_j}, DQ_{O_j})$. Where $LO_j$ is the current position of $O_j$, $DV_{O_j}, DQ_{O_j}$ are two sets, which contain data object $O_j$ sorting all network nodes and query points in ascending order of road network distance within a certain range.

**Definition 4 (node distance matrix, $M$).** The node distance matrix $M$ records the shortest distance between $n$ nodes in the road network $N$, and the matrix $M(V_{i,j})_{n \times n}(i = 1, 2, \ldots, n); j = (1, 2, \ldots, n)$, where $n$ is the total number of road network intersection. As shown in Eq. (1), $V_{1,n}$ is the shortest path distance from node $V_1$ to node $V_n$. The shortest path distance between road network nodes can be stored in $M$ through pre-calculation,

which improves the efficiency of road network distance calculation.

$$\begin{bmatrix} v_{1,1} & \dots & v_{1,n} \\ \dots & \dots & \dots \\ v_{n,1} & \dots & v_{n,n} \end{bmatrix} \tag{1}$$

**Definition 5 (Road network distance, $D$).** The road network distance is based on the shortest path distance of the actual road network, such as road network and railway network. In this article, the road network distance between query point $Q_i$ and data object $O_j$ can be calculated by Eq. (2). When query point $Q_i$ is adjacent to data object $O_j$, the road network distance between them is Euclidean distance $d(Q_i, O_j)$, which is calculated by Eq. (3), where $x_1, y_1, x_2$ and $y_2$ are the longitude and latitude of $Q_i$ and $O_j$. When $Q_i$ and $O_j$ are not adjacent, assume $Q_i$ is on the edge $(v_x, v_y)$. Firstly, the network node $v_p$ and $v_q$ closest to $v_x$ and $v_y$ are found, and then the distances from $v_p$ and $v_q$ to $O_j$ are calculated respectively. Finally, the distances of the two methods are compared, and the smaller distance result is selected as the network distances of $Q_i$ and $O_j$. As shown in Fig. 2, since $Q_5$ and $O_4$ are adjacent, so the road network distance $D(Q_5, O_4) = d(Q_5, O_4)$. Similarly, $Q_6$ and $O_2$ are not adjacent, so $D(Q_6, O_2) = min(d(Q_6, V_{13}) + v_{13,7} + d(v_7, O_2), d(Q_6, V_{14}) + v_{14,8} + d(V_8, O_2))$.

$$\begin{cases} D(Q_i, O_j) & = d(Q_i, O_j) \qquad Q_i \text{ and } O_j \text{ are adjacent} \\ D(Q_i, O_j) & = min(d(Q_i, v_x)) + v_{x,p} + d(v_p, O_j), d(Q_i, v_y) + v_{y,q} + d(v_q, O_j) \\ & Q_i \text{ and } O_j \text{ are not adjacent} \end{cases} \tag{2}$$

$$d(Q_i, O_j) = \sqrt{(y_2 - y_1)^2 + (x_2 - x_1)^2} \tag{3}$$

**Definition 6 (Crowdsourcing matching problem based on road network, CMPRN).** Given a set of query points $Q$ with specific initial locations, a set of requesters $O$, which appear dynamically, and a distance function $D(Q_i, O_j)$ in road network. The CMPRN problem is to find a matching $R$, with minimum total distance $Cost(R) = \sum_{Q_i \in Q, O_j \in O} D(Q_i, O_j)$. And $R$ satisfies the following two constraints:

(1) Real-time constraint: When a task appears, the platform must immediately assign a service provider to the service provider before the next requester arrives.

(2) Invariant constraint: Once a service provider is assigned to a requester, the assignment cannot be revoked.

**Theorem 1. CMPRN is an NP complete problem**.

**Proof**. CMPRN is a Travelling Salesman Problem with Time Window (TSPTW), and the TSPTW problem has been proved to be an NP-complete problem in *Savelsbergh (1985)*. The input of TSPTW is an initial time $t_0$, $N$ vertices {1,2..., n}, where 1 is the starting point, the pairwise distance between vertices is $D'$, and each vertex has a time window $i.w = < e_i, l_i >$, where $l_i \geq e_i \geq t_0$. The TSPTW problem is to determine whether there is a path with a distance less than $D'$ that allows a salesman to travel from vertex 1 to all other vertices at $t_0$ and return to vertex 1 within the corresponding time window. From the description of CMPRN and TSPTW problem, we know that CMPRN is a generalization of the general form of TSPTW problem, so CMPRN is an NP-complete problem.

**Table 1 Symbol definition.**

| Symbol | Description |
| --- | --- |
| $N$ | Road network |
| $V, v_i$ | The network node set, network node |
| $DQ_{v_i}, DO_{v_i}$ | $DQ_{v_i}, DO_{v_i}$ are two sets, which contain road network node $v_i$ sorting all query points and data objects in ascending order of road network distance within a certain range |
| $Q, Q_i$ | The query point set, the query point $Q_i$ |
| $DV_{Q_i}, DO_{Q_i}$ | $DV_{Q_i}, DO_{Q_i}$ are two sets, which contain query point $Q_i$ sorting all network nodes and data objects in ascending order of road network distance within a certain range |
| $O, O_j$ | The data objects set, the data object |
| $DV_{O_j}, DQ_{O_j}$ | $DV_{O_j}, DQ_{O_j}$ are two sets, which contain data object $O_j$ sorting all network nodes and query points in ascending order of road network distance within a certain range |
| $K_1$ | The query radius of PKNN algorithm |
| $K_2$ | The query radius of RKNN algorithm |
| $r$ | The initial radius of the PKNN and RKNN algorithm |
| $\alpha$ | The enlarged radius parameter |
| $M, V_{x,y}$ | The node distance matrix, the shortest path distance from node $V_x$ to node $V_y$ |
| $D(Q_i, O_j)$ | The road network distance between $Q_i$ and $O_j$ |
| $d(Q_i, O_j)$ | The road Euclidean distance between $Q_i$ and $O_j$ |
| $R$ | The matching result set |

The CMPRN problem inherently resembles a greedy problem. In real-world scenarios, task publishers and crowdsourcing workers often expect their requests to be fulfilled immediately. However, in a real-time computing scenario, it is impossible to find a global optimal solution for task initiators and crowdsourcing workers. Therefore, it is necessary to calculate its local optimal solution for the CMPRN problem. Table 1 lists the notations used in this article.

## BIDIRECTIONAL K-NEAREST NEIGHBOR SPATIAL CROWD-SOURCING ALLOCATION PROTOCOL BASED ON EDGE COMPUTING (BKNN-CAP)

In order to obtain the local optimal solution of CMPRN problem, bidirectional k-nearest neighbor spatial crowdsourcing allocation protocol based on edge computing (BKNN-CAP) is proposed in this section. The BKNN-CAP protocol includes spatial crowdsourcing task allocation framework based on edge computing (SCTAFEC), positive k-nearest neighbor spatio-temporal query algorithm (PKNN), and reverse k-nearest neighbor spatio-temporal query algorithm (RKNN).

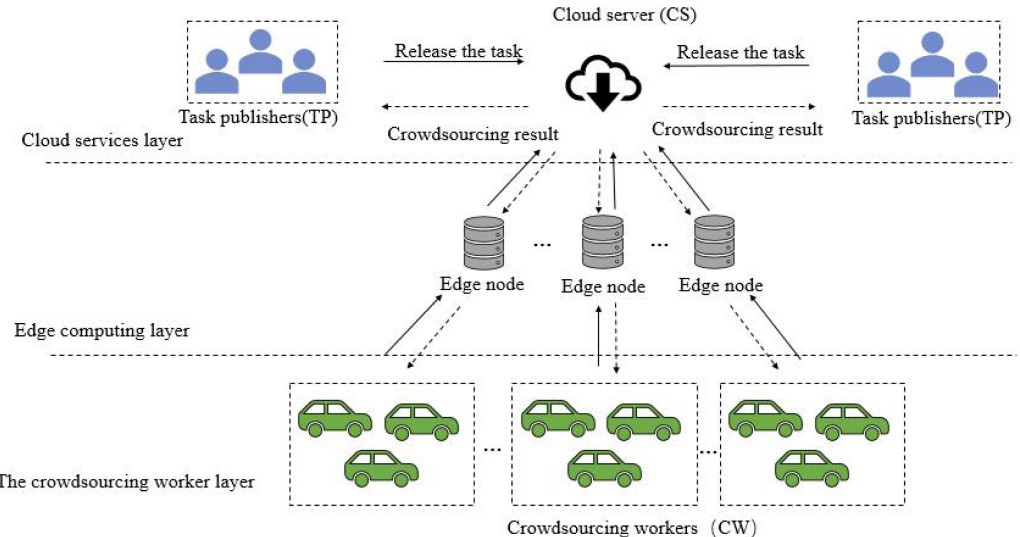

**Figure 3 SCTAFEC system framework.**

## Spatial crowdsourcing task allocation framework based on edge computing (SCTAFEC)

Aiming at the shortcomings of platform framework in subsection "motivation scenario", a spatial crowdsourcing task allocation system framework based on edge computing (SCTAFEC) is proposed. The system framework is shown in Fig. 3, which consists of cloud services layer, edge computing layer and crowdsourcing worker layer.

*Cloud services layer*: it consists of a central control server with high performance and a task publisher (TP) with task publishing requirements. The central control server has a global overview of all available resources and communicates with each deployed edge node server. When the central controller receives task information from the task publisher, the tasks can be offloaded to the edge node server near the task publisher for processing.

*Edge computing layer*: this layer is composed of multiple edge nodes. Each edge node is a server with computing power, which is responsible for processing the crowdsourcing allocation calculation delivered by the central server and returning the results to the central control server. During task distribution, task requests need to be assigned to the nearest edge node for processing based on the specific location of task requests. Since each edge node collects the location information of task publishers and crowdsourcing workers around the node in advance, when the edge node receives the corresponding task request, the reasonable crowdsourcing task allocation will be carried out according to the location information of nearby crowdsourcing workers and task publishers collected by the edge node.

*Crowdsourcing worker layer*: it consists mainly of mobile devices and wireless sensors for crowdsourcing workers (CW). The location information of CW is uploaded to the nearby edge node server through wireless sensor, and the edge node server can allocate the most appropriate crowdsourcing task for CW according to the road network distance.

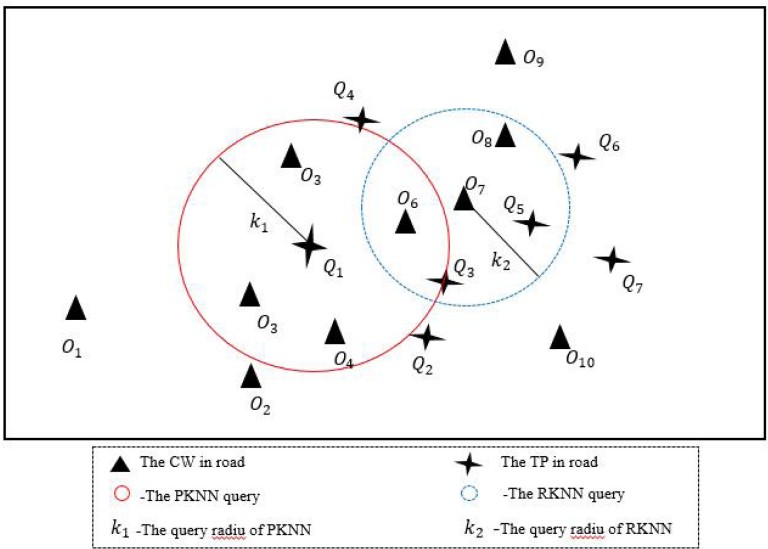

| | |
|---|---|
| ▲ The CW in road | ✦ The TP in road |
| ⭕ -The PKNN query | ⭕ -The RKNN query |
| $k_1$ -The query radiu of PKNN | $k_2$ -The query radiu of RKNN |

**Figure 4** An example of BKNN-CAP protocol.

## Positive k-nearest neighbor spatio-temporal query algorithm (PKNN)

Different from the traditional KNN algorithm, the idea of PKNN algorithm in this section is as follows: given a set of data objects $O$ and a set of query points $Q$, find the data object closest to the query point $Q_i$ road network from the $O$ set within $K$ miles. If the search fails within $K$ miles, expand the query radius $K$ until the search succeeds. As shown in Fig. 4, the circular area with solid red line is PKNN query, where query point $Q_1$ can search for the data object closest to road network within the area with radius $K_1$.

**Step1:** construct the road network node distance matrix $M$, the ascending sequence $DV_{Q_i}$ of the query point to the road network node and the ascending sequence $DO_{v_i}$ of the road network node to the data object.

**Step2:** query stage. Find the road network nodes $v_x$ and $v_y$ in the front and rear directions of the query point within the initial radius $K_1$ miles, and the road network distance of the nearest data object from $Q_i$ to $v_x$ and $v_y$ are calculated respectively. The data object with the smallest road network distance was the matching result of the query point, and the algorithm ends. If no data object can be found within $K_1$ miles, then $K_1 = r + \alpha$ continues (lines 1-8 in Alg. 1), where $\alpha = \alpha + \tau$, when the road network is not dense, the value of $\tau$ can be a larger value. Conversely, when the road network is dense, the value of $\tau$ can be smaller. In this article, $\tau = 1$ is taken as the graph is relatively dense.

**Step3:** in the dynamic update phase, the newly emerged query points and data objects are inserted into the collections $Q$ and $O$, and the successfully matched query points and data objects are deleted from $Q$ and $O$ (lines 9-11 in Alg. 1).

An example of PKNN algorithm is shown in Fig. 5. Firstly, the nearest front and rear road network nodes $v_x$ and $v_y$ from the query point $Q_i$ are foud. Secondly, Search for the latest data objects $O_m$ and $O_n$ of road network nodes $v_x$ and $v_y$. Thirdly, the road network distances of data object $O_m$, $O_n$, and query point $Q_i$ are compared respectively, and the

data object with the closest road network distance is taken as the final matching result. Where $d(Q_i, v_x), d(v_p, O_m), d(Q_i, v_y)$ and $d(v_q, O_n)$ are calculated by Euclidean distance respectively. $v_{xp}$ and $v_{yq}$ can be pre-calculated through A-star algorithm (*Ju, Luo & Yan, 2020*).

The pseudocode of PKNN algorithm is shown in Alg. 1.

---

**Algorithm 1:** Positive K-nearest neighbor spatio-temporal query algorithm (PKNN)

**Input**: The road network $N$, a query points set $Q$, a data object points set $O$, the value of $r, \alpha$.

**Output**: Global task matching result set $R$

1   for $Q_i$ in $Q$ do
2    Initial value $r = 1, \alpha = 0$. Find the nearest road network node $v_x$ and $v_y$ in the front and rear direction of query point $Q_i$ within $K_1$ miles;
3     if $DO_{v_x} = \emptyset, DO_{v_y} = \emptyset$, then
4      $\alpha = \alpha + 1, K_1 = r + \alpha$;
5     else
6      the road network distance of the nearest data object from $Q_i$ to $v_x$ and $v_y$ are calculated respectively. The data object with the smallest road network distance is the matching result of the query point;
7     end if
8   end for
9   for new and departed $Q_a$, $O_b$ do
10    Add new elements and delete those that have left;
11   end for
12   return match result set $R$

---

## Reverse k-nearest neighbor spatio-temporal query algorithm (RKNN)

Due to the PKNN algorithm can only be initiated by query points to search for matching results of query points, it cannot be initiated by data objects to search for matching results of data objects, so the RKNN algorithm is proposed in this section. The RKNN algorithm refers to a given set of data objects $O$ and a set of query points $Q$, this query finds the nearest query point $Q_i$ from the $Q$ set to the data object $O_j$ within $K_2$ miles. As shown in Fig. 4, the blue dotted circle is an RKNN query, where the data object $O_7$ can find the nearest query point of the road network in the area with a radius of $K_2$.

**Step1:** construct the road network node distance matrix $M$, the ascending sequence $DQ_{O_j}$ of the data object to the road network node and the ascending sequence $DQ_{v_i}$ of the road network node to the query point.

**Step2:** query stage. Find the road network node $v_x$ and $v_y$ in the front and rear directions of the data object within the initial radius $K_2$ miles, and the road network distance of the nearest query point from $O_j$ to $v_x$ and $v_y$ are calculated respectively. The query point with the smallest road network distance was the matching result of the data object, and the algorithm ends. If no query point can be found within $K_2$ miles, then $K_2 = r + \alpha$ continues, where $\alpha$ is the enlarged radius parameter (lines 1-8 in Alg. 2).

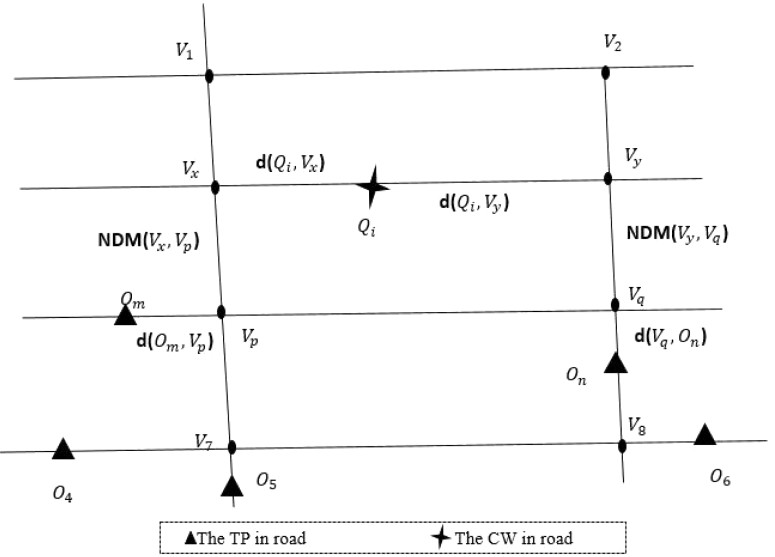

**Figure 5** An example of PKNN query calculation.

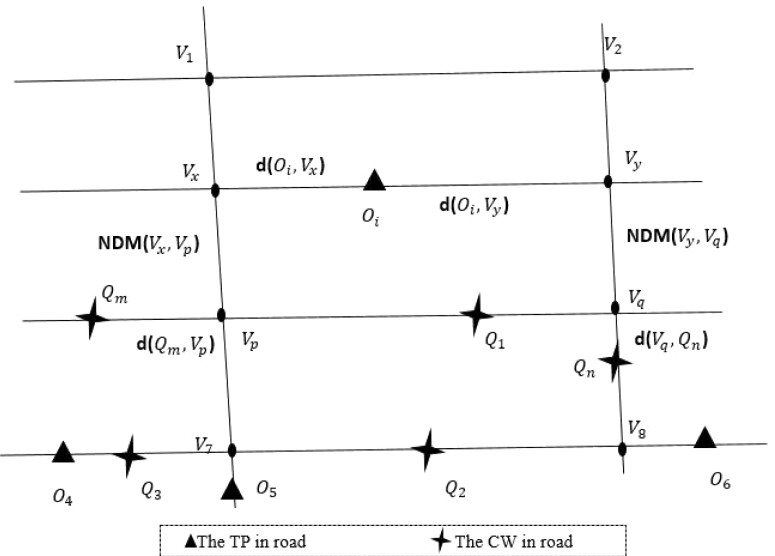

**Figure 6** An example of RKNN query calculation.

**Step3:** in the dynamic update phase, the newly emerged query points and data objects are inserted into the collections $Q$ and $O$, and the successfully matched query points and data objects are deleted from $Q$ and $O$ (lines 9-11 in Alg. 2).

Figure 6 shows an example of the RKNN algorithm. When the data object $O_i$ initiates a query, the following processing is performed. Firstly, find the nearest left and right road network nodes $v_x$ and $v_y$ from the data object $O_i$. Secondly, search for the latest query point $Q_m$ and $Q_n$ of road network node $v_x$ and $v_y$. Thirdly, the road network distances of query

point $Q_m, Q_n$ and data object $O_i$ are compared respectively, and the query point with the closest road network distance is taken as the final matching result.

The pseudocode of RKNN algorithm is shown in Alg. 2.

---

**Algorithm 2:** Reverse K-nearest neighbor spatio-temporal query algorithm (RKNN)

---

**Input**: The road network $N$, a query points set $Q$, a data object points set $O$, the value of $r, \alpha$.

**Output**: Global task matching result set $R$

1  for $O_i$ in $O$ do
2    Initial value $r = 1, \alpha = 0$. Find the nearest road network node $v_x$ and $v_y$ in the front and rear direction of data object $O_i$ within $K_2$ miles;
3      if $DQ_{v_x} = \varnothing, DQ_{v_y} = \varnothing$, then
4        $\alpha = \alpha + 1, K_2 = r + \alpha$;
5      else
6        the road network distance of the nearest data object from $O_i$ to $v_x$ and $v_y$ are calculated respectively. The query point with the smallest road network distance was the matching result of the data object;
7      end if
8    end for
9    for new and departed $Q_a$, $O_b$ do
10      Add new elements and delete those that have left;
11   end for
12   Return match result set $R$

---

## SIMULATION EXPERIMENT ANALYSIS

The spatial crowdsourcing scenario used in this article is an online car-hailing system, in which passengers are the query points in the spatial crowdsourcing platform and the drivers are the data objects. In traditional online taxi-hailing system, all computing processing is carried out on the central server. In this article, the edge computing layer is introduced. The central control server is responsible for distributing computing tasks to the edge nodes for processing, thus reducing the computing pressure on the central server.

### Simulation environment and data set

In this article, the simulation experiment environment of the central control server is 64-bit Windows 10 system, memory (RAM) is 8.00GB, and processor is Intel (R) Core I7. Experimental data are mainly derived from partial road network data of Fuzhou captured from OpenStreetMap. The original network data set contains 16,453 road nodes and 17,801 roads composed of these road nodes. When the original data was used in the experiment, there was too much data and redundant data. Therefore, in order to ensure the effectiveness of the experiment and better verify the model, we optimized the selection of the original road network data and simplified it to a certain extent. In the case of multiple small segments in a road, roads were merged, which greatly reduced the road node set, road network crossing node set and road network boundary set. The final data set processed

included 4,116 road nodes, 3,457 road network crossing nodes and 5464 roads. Assume that there are 3,000 base stations in the network. The coverage radius of the base stations is 500 m. The CPU of the edge node server is RK3399 and the Cortex A53 quad-core 1.4 GHz and A72 dual-core 2 GHz. The main frequency is up to 2.0 GHz, the memory is 2 GB, the storage is 4 GB, and the communication channel bandwidth is 1 MHz. Such a parameter setting can enable each edge node server to collect the location information of ride-hailing and passengers within a radius of 500 m, and to process calculation tasks with moderate data volume. Figure 7 shows the road network after final treatment.

## Feasibility analysis

As shown in Fig. 8, U64 is a passenger ID 64, and 8 is a driver ID 8. U64 and 8 are matched successfully after the test run, and the blue route is the network route from driver to passenger. Drivers can connect to U64 along this road. This proves the feasibility of the BKNN-CAP protocol proposed in this article.

## Evaluation indicators

Five evaluation indicators are introduced in this section, namely time cost, average response time, energy consumption cost, the number of successful matches and matching success rate.

(1) Time cost

The time cost $T$ in this article refers to the time cost required to complete the matching of $M$ passengers and $N$ task initiators at a certain time. Its calculation is obtained from Eq. (4), where $t_1$ is the pre-calculated time of the protocol and $t_2$ is the query time of the protocol.

$$T = t_1 + t_2 \tag{4}$$

(2) Average response time

Average response time $T_a$ refers to the time taken by each passenger in the spatial crowdsourcing platform from the initiation of a taxi request to the successful matching of online car-hailing. The shorter the average response time $T_a$, the better the service quality of the platform. In this article, the average response time $T_a$ is calculated by Eq. (5). Where $n$ is the number of passengers in the spatial crowdsourcing platform at a certain moment.

$$T_a = T/n \tag{5}$$

(3) Energy consumption cost

Energy consumption cost refers to the energy cost generated by the scheme matching results in practical application, such as the electricity cost of electric vehicles or the gasoline cost of fuel vehicles. Since the cost of energy consumption is directly proportional to the journey of an online car, in order to conveniently calculate the cost of energy consumption, the road network distance of the online car-hailing is used to represent the energy consumption cost in this article.

(4) The number of successful matches

In the simulation experiment, due to some objective factors, such as the location of passengers is too remote and other factors, the nearby online taxi cannot be matched.

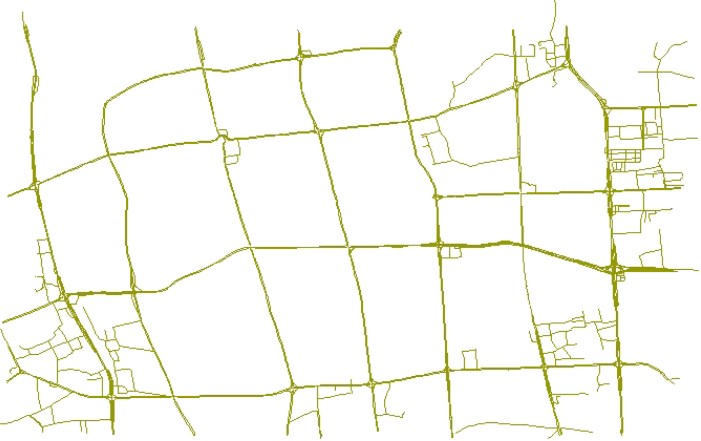

**Figure 7** **An example of some road networks in Fuzhou.**

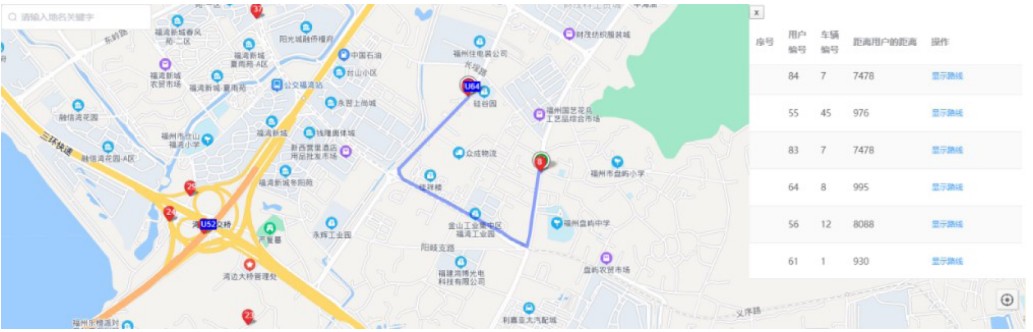

**Figure 8** **An example of driver passenger matching visualization.** Map source credit: ©2022 Baidu.

Therefore, the number of successful matching in this article refers to the number of passengers successfully matched to the nearby online taxi in a certain period of time. The higher the number of successful matches, the better the service quality of the spatial crowdsourcing platform.

(5) Matching success rate

Matching success rate (MSR) refers to a certain time period the proportion of passengers to successful matching to the nearby mesh about, its computation as shown in Eq. (6), where $|O|$ is the number of drivers, $|R|$ indicates the number of successful matches.

$$MSR = |R|/|O|. \qquad (6)$$

## Comparative analysis

In view of the above indicators, the BKNN-CAP protocol proposed in this article will be compared and analyzed with the CAKNN algorithm in *Miao et al. (2020)* and pRMatch algorithm in *Yu, Zhang & Yu (2020)*.

(1) Comparative analysis of time cost

As shown in Fig. 9, the time consumption of the BKNN-CAP protocol proposed in this article is lower than that of CAKNN algorithm and pRMatch algorithm. The reason is that BKNN-CAP protocol adopts the idea of edge computing and offloads all task requests to edge nodes for processing, which greatly improves the efficiency of crowdsourcing allocation. For example, when the volume of online car booking is 200, the time consumed by BKNN-CAP, pRMatch and CAKNN algorithms is 8,451, 15,048 and 39,608 ms, respectively. In addition, the time consumption of BKNN-CAP protocol is still lower than that of the other two algorithms when the number of drivers keeps increasing, which reflects the superiority of the algorithm proposed in this article in terms of time.

(2) Comparative analysis of average response time

As shown in Fig. 10, the average response time of both CAKNN algorithm and pRMatch algorithm increases with the increase of the driver scale. When the driver scale is less than 150, the average response time of BKNN-CAP is slightly higher than that of pRMatch, because BKNN-CAP needs to conduct the prediction calculation of the node distance matrix, while pRMatch has no prediction calculation. When the size of the driver is larger than 150, BKNN-CAP protocol avoids double calculation due to the precalculation, so that the average response time tends to be stable, and the duration is below 70 ms, which greatly reduces the query waiting time of passengers and drivers in the spatial crowdsourcing platform, thus improving the user experience of the spatial crowdsourcing platform.

(3) Comparative analysis of energy consumption cost

As shown in Fig. 11, the energy consumption of the BKNN-CAP protocol proposed in this article is lower than that of CKNN and pRMatch algorithms. For example, when the volume of online ride-hailing car is 200, the energy cost consumed by BKNN-CAP, pRMatch and CAKNN algorithm is 2012, 2463 and 2198 respectively. When the driver scale is less than 150, some CAKNN points are higher than pRMatch, but some are not. The reason is that CKNN algorithm is accurate in calculating road network distance, while pRmatch algorithm is obtained by approximate calculation. When the size of the driver is small, the energy consumption between the two is not stable, and its superiority cannot be better reflected. However, when the driver scale is larger than 150, CAKNN calculates the road network distance more accurately, so the road network distance is shorter, so the response energy consumption is lower than pRmatch. On the whole, the BKNN-CAP protocol is more energy efficient than the other two algorithms, because the bidirectional k-nearest space–time query mechanism can find the real nearest ride-hailing driver for each passenger, thus making the network distance shorter.

(4) Comparative analysis of successful matching quantity

As shown in Fig. 12, when the size of drivers is 50, the matching success amounts of BKNN-CAP, CAKNN and pRMatch algorithms are 24, 22 and 25, respectively. However, when the driver size increases, the matching success of BKNN-CAP query protocol is higher than that of the other two algorithms. For example, when the driver size is 300, the matching success of BKNN-CAP, CAKNN and pRMatch algorithm are 224, 193 and 206, respectively. In practical application scenarios, the number of drivers on the crowdsourcing platform is generally more than 50. Therefore, the BKNN-CAP protocol proposed in this

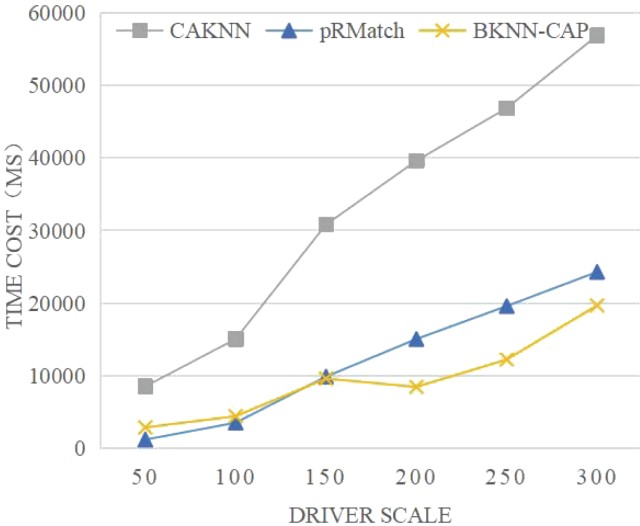

**Figure 9  Comparison of average response time of algorithms.**

article is more practical. Some CAKNN points are higher than pRMatch but some are not, because when the driver scale is less than 200, the calculation result of road network distance of CAKNN algorithm is more accurate, so its successful matching quantity is higher than pRMatch. When the driver scale is greater than 200, the road network distance of pRMatch algorithm adopts the approximate calculation method. Therefore, the crowdsourcing matching efficiency of pRMatch algorithm is higher, so the number of successful matching is higher than that of CAKNN algorithm.

(5) Analysis of matching success rate

As shown in Fig. 13, the MSR of BKNN-CAP protocol is always higher than pRMatch and CAKNN algorithm. This is because the BKNN-CAP protocol is a two-way matching query mechanism, while the other two algorithms are one-way queries. When the volume of online ride-hailing cars is 200, the MSR of BKNN-CAP, pRMatch and CAKNN algorithms are 69%, 53% and 56%, respectively. In addition, with the continuous expansion of drivers, the matching success rate of the BKNN-CAP protocol proposed in this article keeps rising, while the matching success rate of CAKNN and pRMatch algorithms shows volatility. Therefore, the BKNN-CAP protocol is more stable.

## CONCLUSIONS

In spatial crowdsourcing platform, how to provide intelligent services while ensuring a high quality of service is an important yet challenging problem. To solve this problem, the bidirectional k-nearest neighbor spatial crowdsourcing allocation protocol based on edge computing (BKNN-CAP) is proposed in this article. Firstly, a spatial crowdsourcing task allocation system framework based on edge computing (SCTAFEC) is established. With this framework, all tasks can be unloaded to the edge nodes of the edge computing layer for processing, which greatly reduces the service response time. Secondly, based on the KNN

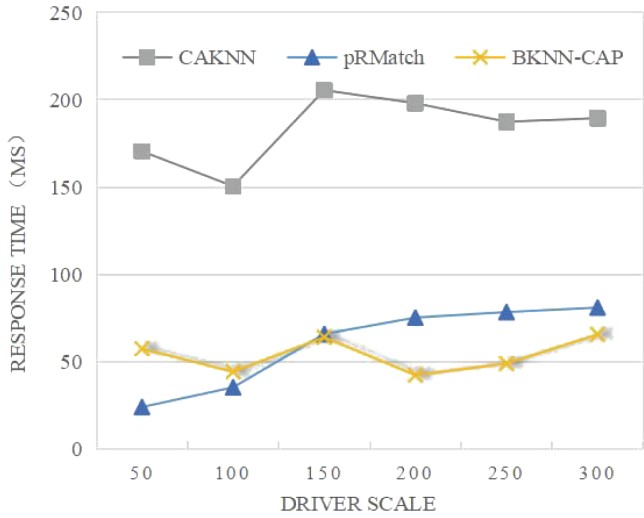

**Figure 10** Comparison of average response time of algorithms.

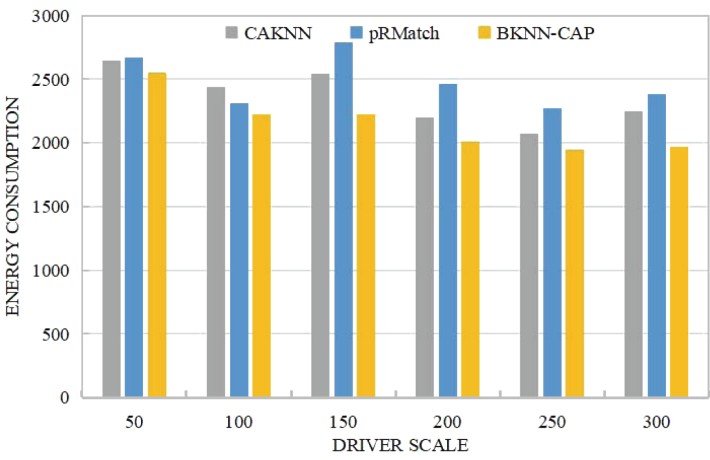

**Figure 11** Comparison of energy consumption.

algorithm, the positive k-nearest neighbor spatio-temporal query algorithm (PKNN) and reverse k-nearest neighbor spatio-temporal query algorithm (RKNN) are proposed, which make query points and data objects query in parallel. Thirdly, in order to further improve the matching accuracy, a road network distance calculation method is proposed based on the urban road network. Finally, sufficient experiments are carried out on real road network data sets to prove the superiority of BKNN-CAP protocol in time cost, energy consumption cost, matching success and other indicators.

Future studies in this research can be performed in the following directions. Firstly, location privacy protection technology will be applied to BKNN-CAP protocol to achieve crowdsourcing matching and protect users' location privacy data at the same time. Secondly,

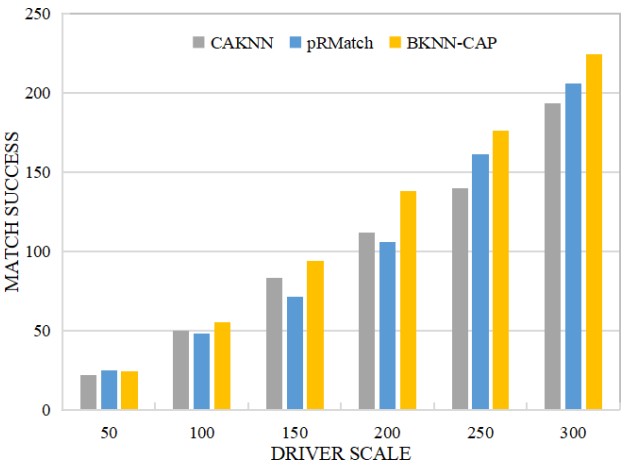

**Figure 12 Comparison of successful matching quantity.**

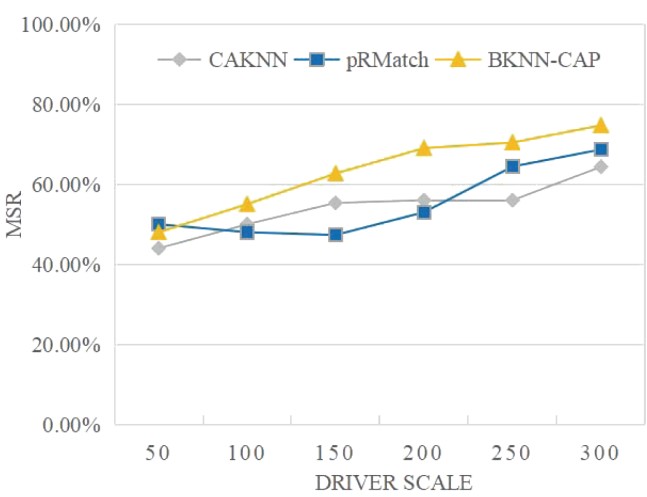

**Figure 13 Comparison of matching success rates.**

the combination of BKNN-CAP protocol with other optimization objectives will be studied.

### Funding

This work is funded by the National Natural Science Foundation of China (No. 61902069 and the U1905211), Natural Science Foundation of Fujian Province of China (2021J011068). The funders had no role in study design, data collection and analysis, decision to publish, or preparation of the manuscript.

### Grant Disclosures

The following grant information was disclosed by the authors:
The National Natural Science Foundation of China: No. 61902069, U1905211.
Natural Science Foundation of Fujian Province of China: 2021J011068.

### Competing Interests

The authors declare that they have no competing interests.

### Author Contributions

- Jing Zhang conceived and designed the experiments, analyzed the data, prepared figures and/or tables, authored or reviewed drafts of the article, and approved the final draft.
- Qian Ding conceived and designed the experiments, analyzed the data, prepared figures and/or tables, authored or reviewed drafts of the article, and approved the final draft.
- Biao Li performed the experiments, performed the computation work, prepared figures and/or tables, authored or reviewed drafts of the article, and approved the final draft.
- Xiucai Ye performed the experiments, performed the computation work, authored or reviewed drafts of the article, and approved the final draft.

### Data Deposition

The original data set, the code, and the data used to generate the figures are in the Supplementary Files.

### Supplemental Information

Supplemental information for this article can be found online at http://dx.doi.org/10.7717/peerj-cs.1244#supplemental-information.

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
