# Peer review of "Bidirectional k-nearest neighbor spatial crowdsourcing allocation protocol based on edge computing"

_PeerJ Computer Science, doi:10.7717/peerj-cs.1244_

## Round 0.1 · original submission · Major Revisions

Please address the comments from the reviewers in preparing your revision.

Reviewer 1 ·

Basic reporting

no comment

Experimental design

1. The paper is about the research of the k-nearest neighbor algorithm with bi-directional exploration starting from the source and destination separately. Their work is about increasing the algorithm performance in an edge computing environment.

2. They clearly define the motivation and the problem. They also provide the source of their data, how they preprocess their data, and give some illustrations about how their abstract their problem. They also provide pseudo codes for their algorithm.

3. They provide a metric with time, energy cost, response time, and accuracy to evaluate their algorithm compared to their previous work.

4. They provide the source data and algorithm code, so I believe they supply sufficient details and information to replicate.

Validity of the findings

I do not rerun their experiments, but based on the algorithm logic, the run time should be better than one directional algorithm.
The data is provided and their conclusion is concise.

Additional comments

Several tech questions:
1. They use only one city data as the test. I understand that the data retrieval and preprocessing takes some time but only one city is not illustrative enough to persuade readers that the algorithm is good enough. Additionally, their graph consists of roughly 3000 points and 3000 edges, that's not a very big graph. It's not clear enough to show that distributing several edge computing nodes is necessary.

2. How they should define the alpha if the graph is not dense?

3. They should have a clear definition of the computing power of edge computing. If their nodes are running windows with intel CPU and 8GB, it looks more like an obsolete data center node. For typical edge computing, it should have even less computing power.

4. When you distribute the workloads, do you also send initial data to several nodes? (Each node has a complete initial copy of data and there is no request for data transfer when the nodes are running) The workloads can just be processed in parallel and there is no inter-task dependency.

5. Concerning novelty, this algorithm idea is some sort like bi-directional BFS for a path from source to destination. I still agree that the approach can be applied to their problem and their work is valid. But I think they can still defend for themselves.

·

Basic reporting

This article proposes an efficient distributed and bidirectional protocol for crowdsourcing allocation systems. The authors identify three problems in the existing approaches: 1) low processing efficiency because of the centralised framework; 2) unreversable query because of one-way allocation; and 3) inaccuracy because of distance calculation using Euclidean distance. This article tackles these three problems by proposing a new protocol.

Experimental design

The problems are well-motivated and explained well.
The motivation example is interesting. However, the formulation section needs improvements.
- First, the authors introduce a few definitions but need more explanations on their elements. For instance, I know N is a 3-tuple model but the dimension of E and W are not explained. What are weights, and why are they needed? DQ and DO also need more clarification. I suggest the authors to use examples from Fig. 3.
- P5l185: n is not defined.
- Eq. 2 needs more explanation. x, p, y and q are not defined.
- Tab. 1 has wrong definitions. For instance, is DV an order? Or it is a vector with elements in ascending order? On page 5, it is called “set”, which does not have order information. Please clarify.

The difference between the proposed framework and the baseline framework is a bit confusing. In Fig. 4, what if I consider the cloud server and all the edge computing layer as a single system node? What are the difference between this and Fig. 1? Please clarify.

Validity of the findings

The results look promising. The evaluation parameters are listed and explained. However, more interpretation is needed. The authors describe their observation but do not explain why this makes sense. For instance, why BKNN-CAP is higher than prematch in Fig. 11 and Fig. 14? Why some CAKNN points are higher than pRMatch but some are not in Fig. 13 and Fig. 12? Please clarify.

Additional comments

Small issues:
p1l30: “According to Marketsand Markets,” => citation missing
p2l97: “total utility value of” => This term is not mentioned anywhere else. Please give a definition
p3l147: ”Motivation scenario” => ``Motivation scenario’’ Same for the rest of quotes.
p5l190: “fig.2” => “fig.3”?
Fig. 12 and Fig. 13: “PRMATCH” => “pRMatch”?

---

## Round 0.2 · accepted · Accept

The authors have addressed the concerns of the reviewers and the paper is ready for publication.

·

Basic reporting

no comment

Experimental design

no comment

Validity of the findings

no comment

Additional comments

I am happy with the significant revisions presented in this manuscript. Thanks for the effort.